# Mega macromolecules as single molecule lubricants for hard and soft surfaces

Parambath Anilkumar [1,2,11], Taylor B. Lawson [3,4,11], Srinivas Abbina [1,2,11], Janne T. A. Mäkelä [4,5,6,11], Robert C. Sabatelle[6], Lily E. Takeuchi [1,2], Brian D. Snyder[4,6], Mark W. Grinstaff [3,6,7,8,12] & Jayachandran N. Kizhakkedathu [1,2,9,10,12]

A longstanding goal in science and engineering is to mimic the size, structure, and functionality present in biology with synthetic analogs. Today, synthetic globular polymers of several million molecular weight are unknown, and, yet, these structures are expected to exhibit unanticipated properties due to their size, compactness, and low inter-chain interactions. Here we report the gram-scale synthesis of dendritic polymers, mega hyperbranched polyglycerols (mega HPGs), in million daltons. The mega HPGs are highly water soluble, soft, nanometer-scale single polymer particles that exhibit low intrinsic viscosities. Further, the mega HPGs are lubricants acting as interposed single molecule ball bearings to reduce the coefficient of friction between both hard and soft natural surfaces in a size dependent manner. We attribute this result to their globular and single particle nature together with its exceptional hydration. Collectively, these results set the stage for new opportunities in the design, synthesis, and evaluation of mega polymers.

[1] Centre for Blood Research, Life Sciences Institute, The University of British Columbia, Vancouver, BC, Canada. [2] Department of Pathology and Laboratory Medicine, The University of British Columbia, Vancouver, BC, Canada. [3] Department of Mechanical Engineering, Boston University, Boston, MA, USA. [4] Center for Advanced Orthopaedic Studies, Beth Israel Deaconess Medical Center, Harvard Medical School, Boston, MA, USA. [5] Biophysics of Bone and Cartilage, Department of Applied Physics, University of Eastern Finland, Kuopio, Finland. [6] Department of Biomedical Engineering, Boston University, Boston, MA, USA. [7] Department of Medicine, Boston University, Boston, MA, USA. [8] Department of Chemistry, Boston University, Boston, MA, USA. [9] Department of Chemistry, The University of British Columbia, Vancouver, BC, Canada. [10] School of Biomedical Engineering, The University of British Columbia, Vancouver, BC, Canada. [11] These authors contributed equally: Parambath Anilkumar, Taylor B. Lawson, Srinivas Abbina, Janne T. A. Mäkelä. [12] These authors jointly supervised this work: Mark W. Grinstaff, Jayachandran N. Kizhakkedathu. ✉email: mgrin@bu.edu; jay@pathology.ubc.ca

As our mastery of polymerization reactions advances, macromolecules of increasing complexity and size are readily synthesized. Dendrimers clearly showcase this advancement; specific structural or compositional features are introduced at defined locations within a 3D globular structure[1–6]. Linear polymers of a few million dalton (MDa) molecular weight also exhibit interesting properties, and are prepared via judicious choice of monomer and under carefully controlled reaction conditions, including reversible deactivation radical polymerization[7], atom transfer radical polymerization[8], reversible addition-fragmentation chain transfer[9], ring-opening metathesis[10], and Lewis pair polymerizations[11]. Synthesizing globular polymers of several million molecular weight has not been previously achieved and will, similarly, afford unexpected properties as one single polymer itself occupies a size of tens of nanometers owing to its compact structure. A path to such mega-macromolecules of several MDa molecular weight represents a significant synthetic challenge. During a polymerization reaction, the viscosity of the polymerization medium increases, which leads to the quenching of propagation species, poor control of molecular weight, polydispersity, and branching.

Herein, we report the synthesis and characterization of semidendritic hyperbranched polyglycerols of 1, 3, and 9 MDa (i.e., mega HPGs), and their performance as single molecule soft ball bearing lubricants on hard and soft surfaces. The mega HPGs reduce the coefficient of friction (COF) between both hard and soft surfaces, and exhibit rheological properties similar to a fluid lubricant. The high water solubility, low intrinsic viscosity, compactness, nearly molecular weight independent intrinsic viscosity, and hydration of the mega HPGs are responsible for their performance as single molecule ball bearings.

## Results

### Synthesis and structural characterization of mega HPGs.

To overcome the challenges and prepare a large globular polymer, we used a homogenous polymerization method along with a macroinitiator to obtain ultra large dendritic polymers possessing a globular shape. As our interests lie in the polymers of potential use for environmentally responsible high-performance and biomedical applications, the selection of building blocks that are degradable, green, biocompatible, or natural metabolites is an additional key design criterion[12,13]. Polymers possessing a glycerol backbone are of significant interest for biomedical and environmental-green applications due to their chemical tuneability, degradability, and biocompatibility[14–21]. We synthesized the mega HPGs, in the MDa range (up to 10 MDa), via ring-opening multibranching polymerization (ROMBP) in a single pot, using a combined macroinitiator and solvent-based solution polymerization approach (Fig. 1)[22]. Specifically, we used the partially deprotonated (10%) high molecular weight HPG (KH in DMF, $M_w$ 840 kDa, Đ 1.2) as the macroinitiator (Supplementary Figs. 1–3). The high solubility of the deprotonated macroinitiator in the polymerization medium, dry conditions, and the use of KH as deprotonating base are critical to afford uniform polymer growth from the macroinitiator. The slow addition of the monomer glycidol at 95 °C produced predetermined molecular weights of mega HPGs in a controlled manner in good yields (Fig. 1a, Table 1). We synthesized three different molecular weight mega HPGs (mega HPG-1 ($M_w$ 1.3 MDa, Đ 1.2), mega HPG-2 ($M_w$ 2.9 MDa, Đ 1.2), and mega HPG-3 ($M_w$ 9.3 MDa, Đ 1.4)) by changing the glycidol to macroinitiator ratio (Table 1). The mega HPGs exhibit a monomodal distribution as demonstrated using gel permeation chromatography with multi angle light scattering (GPC-MALS; Fig. 1d, Table 1, Supplementary Figs. 4 and 5). Homogeneous polymerization conditions in dry solvent and the

maintenance of uniform stirring using an overhead stir are important determinants in synthesizing mega HPGs with low polydispersity in good yield (~74–85%). For example, the current protocol affords ~42 g of isolated mega HPG-3 in a single batch. We repeated all of the reactions at least twice and obtained similar results (for example, average $M_w$ for mega HPG-3 from three different batches was 9.3 ± 0.03 MDa and the yield was 74 ± 0.21%). This is significant, considering the difficulty in synthesizing semidendritic/hyperbranched polymers of such high molecular weights in large quantities, using anionic ring-opening polymerization.

We further characterized the mega HPGs by nuclear magnetic resonance (NMR) analysis for structure and branching density (Fig. 1, Supplementary Figs. 6–9). A representative NMR spectra of mega HPG-3 shows the characteristic peaks of the polyglycerol backbone (Fig. 1b, c)[23]. The degree of branching of mega HPGs, determined by $^{13}C$ inverse-gated (IG) NMR spectroscopy, is between 53 and 57%, and similar to HPGs reported previously (Fig. 1c, Table 1, Supplementary Table 1) prepared by the ROMBP method[24]. The molecular weights achieved are the highest reported for dendritic polymers and as such these polymers display unique characteristics. For example, mega HPG-3 possesses >87,000 hydroxyl groups per polymer on average influencing properties and amenable to modification. The mega HPGs are highly hydrated, as determined by differential scanning calorimetry (DSC); mega HPG-3 possesses ~389,300 bound water molecules per polymer (Supplementary Fig. 10, Supplementary Table 1). As suspected, the mega HPGs are highly water soluble (>380 mg/mL) in contrast to similarly sized linear polymers (e.g., polyethylene oxide (PEO) and polyvinyl alcohol (PVA)), which form gels at high concentrations (Supplementary Table 2)[25].

Mega HPGs are highly compact nanostructures as demonstrated by their hydrodynamic diameters (determined by dynamic light scattering (DLS)) that range from 21 nm for the 1.3 MDa to 43 nm for the 9.3 MDa polymer (Table 1). We further investigated the size of mega HPGs using cryogenic scanning electron microscopy (cryo-SEM). A representative cryo-SEM of mega HPG-3 shows the spherical and single-particle nature of the mega HPGs (Fig. 1e). The average size of the mega HPGs are 28, 34, and 51 nm for mega HPG-1, 2, and 3, respectively, and slightly higher than the data obtained from DLS likely due to the frozen hydrated shell of the mega HPG in the cryo-SEM preparations (Supplementary Fig. 11). Importantly, individual mega HPGs are nanometer-scale singular polymer particles and easily visualized. The low hydrodynamic sizes of mega HPGs confirms their stability in water without aggregation[26,27], and the size does not scale with molecular weight as with linear polymers[28] and low molecular weight HPG (Fig. 2a, Supplementary Table 1). The size of the mega HPGs lies between linear polymers (e.g., PEG) and dendrimers (e.g., PAMAM; Fig. 2a). The data further indicate that mega HPGs are more compact than linear polymers and less compact in comparison to dendrimers, and this might be advantageous in terms of offering more 'interior room' between branching units.

The mega HPGs possess very low intrinsic viscosity in comparison to other polymeric systems. The intrinsic viscosity, [η], of the mega HPGs does not change considerably with molecular weight (Table 1); this small change in intrinsic viscosity of mega HPGs also does not follow the Mark–Houwink–Sakurada equation and instead falls in line with the Einstein viscosity theory's prediction of almost minimal change in [η] with molecular weight for "hard" globular-shaped polymers[27]. This result is in striking contrast to dendrimers (PAMAM, generation G1–G10), which show an unusual bell shape dependence of [η] with increasing molecular weight (Fig. 2b)[29]. Published

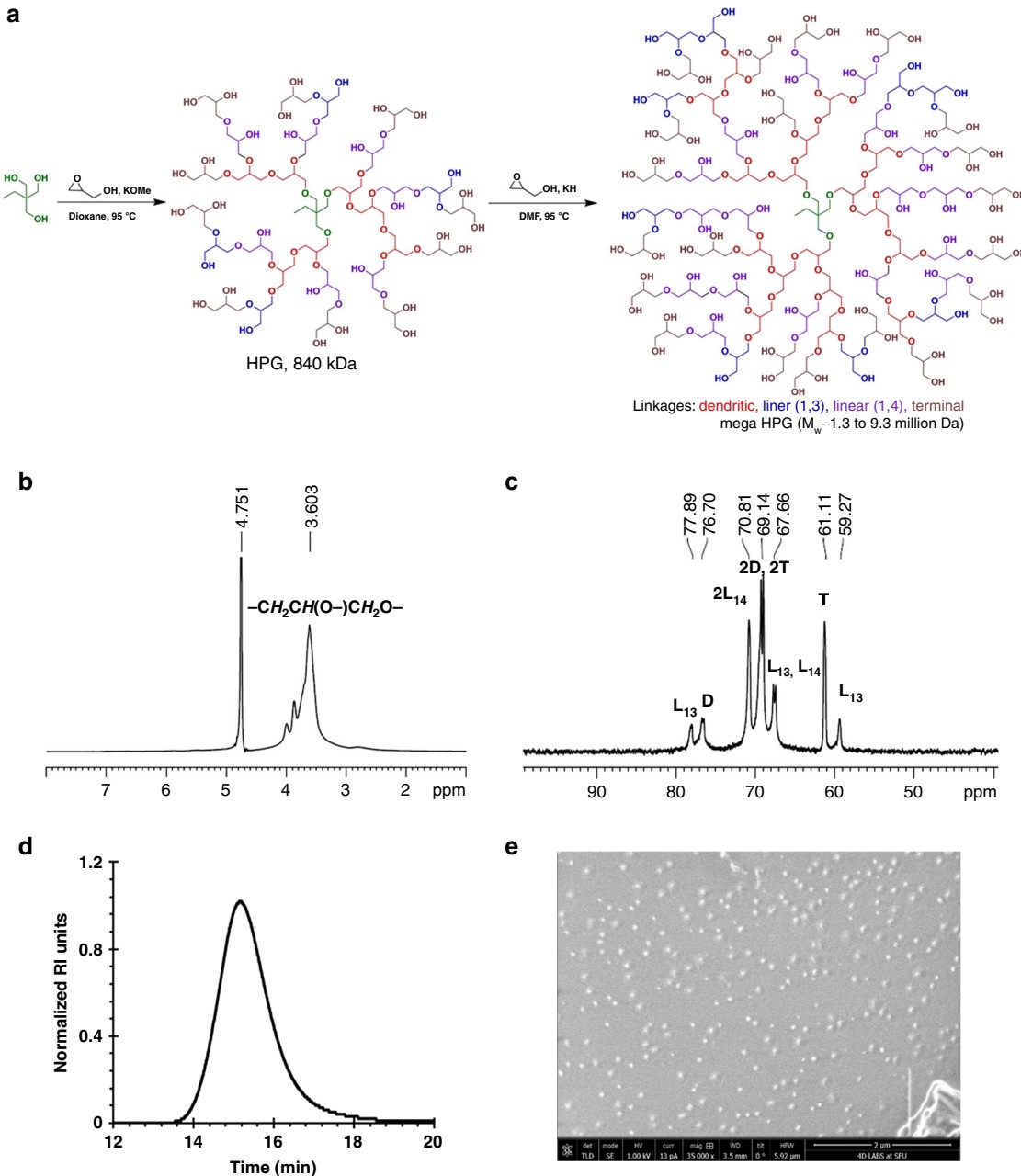

**Fig. 1 Synthesis and characterization of mega HPGs. a** Schematic representation of synthesis of mega HPGs ($M_w$: 1.3–9.3 MDa) by a macroinitiator approach in combination with solvent-based ring-opening multibranching polymerization. **b** $^1$H NMR and **c** $^{13}$C IG NMR characterization confirmed the structural features of the mega HPG-3. **d** Gel permeation chromatography analysis shows the monomodal distribution of mega HPG-3. **e** Formation of single particles and globular shape of mega HPG-3 was confirmed by cryo-SEM.

| Table 1 Physical characteristics of mega HPGs. | | | | | |
|---|---|---|---|---|---|
| Entry | Polymer | $M_w$ (Da) | Đ | DOB | Size (nm) | [$\eta$] (mL/g) |
| 1 | Mega HPG-1 | $1.3 \times 10^6$ | 1.2 | 0.57 | 21.2 ± 0.4 | 4.67 |
| 2 | Mega HPG-2 | $2.9 \times 10^6$ | 1.2 | 0.54 | 30.6 ± 0.6 | 5.26 |
| 3 | Mega HPG-3 | $9.3 \times 10^6$ | 1.4 | 0.53 | 43.0 ± 0.4 | 6.15 |

Absolute molecular weight ($M_w$) and distribution (Đ) of mega HPGs was confirmed by size-exclusion chromatography coupled with light scattering detector (MALS). Degree of branching (DOB—53-57%) supports the semidendritic nature of the polymers and it was determined by $^{13}$C IG NMR spectroscopy. Mega HPGs are compact in size and have low intrinsic viscosity [$\eta$], determined by quasielastic light scattering (QELS) detector and viscometer-II detector, respectively, which are coupled to a gel permeation chromatography system.

 

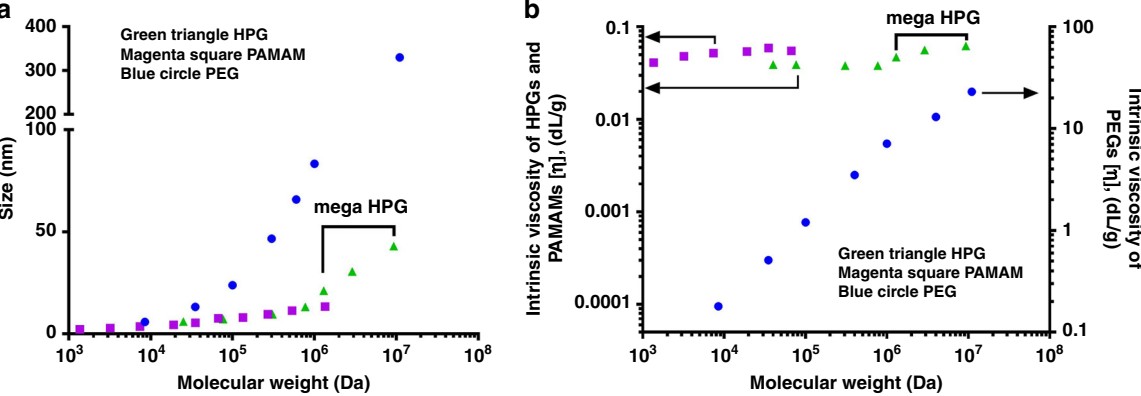

**Fig. 2 Comparison of solution properties of mega HPGs with PEG and PAMAM dendrimers. a** Variation of hydrodynamic size of the polymer with molecular weight. Mega HPGs and their low molecular weight counterparts (first four data points), and PAMAM dendrimers are compact in size compared to PEG polymers. The values for high molecular weight HPGs (76.5, 307, and 771 kDa) are obtained from literature[64]. For PEGs, the size of PEG-4 (11 MDa) was derived from $R_g$ and simulation studies[63]. **b** Dependence of intrinsic viscosity of the polymers with molecular weight (arrow shows the representative y-axis). Mega HPGs showed similar intrinsic viscosity behavior as that of PAMAM dendrimers; however, slight increment with molecular weight might was observed. The PEG systems showed liner dependency with molecular weight.

simulation and experimental studies validate that the PAMAM dendrimer has almost zero asphericity with increasing generation number (G1→9) which indicates substantial back folding of terminal groups into the interior core of the dendrimer structure[30]. This conformational change may be occurring with mega HPGs to some degree as evident from their nearly molecular weight independent intrinsic viscosity; however, additional studies are needed to confirm this notion. Interestingly, the intrinsic viscosities of the mega HPGs are slightly increased with molecular weight compared to their lower molecular counterparts; the increase in size of the mega HPGs might be attributing to this rise[31]. The intrinsic viscosity values of the mega HPGs are significantly lower than linear water-soluble polymers of similar molecular weight (e.g., PEO, hyaluronic acid, and dextran sulfate)[25,32–35], and this observation further confirms the compact nature of the mega HPGs. For example, the mega HPG-3 (9.3 MDa) possesses an intrinsic viscosity of 6.15 mL/g that is similar to most of the globular proteins in aqueous solutions[36–38], whereas a linear PEG ($M_w$ 11 MDa) has a value of 2600 mL/g[28]. Thus, mega HPGs show a unique combination of ultra-high molecular weight, compact size, and very low intrinsic viscosity almost independent of molecular weight.

**Lubrication properties**. To take advantage of the unique solution properties of mega HPGs (high solubility, compactness, and low intrinsic viscosity), we investigated the lubrication properties of mega HPGs on hard synthetic and soft natural surfaces. Considering the globular and single-particle nature of the mega HPGs, we anticipated that the rheological properties of mega HPGs will be quite different from their linear counterparts. We first determined the performance and lubrication characteristics of mega HPGs on stainless steel by generating Stribeck curves. Stribeck curves describe the COF of a system across different lubrication modes: boundary, mixed, and hydrodynamic lubrication (see Supplementary Discussion for details). Generally, boundary lubrication exists under conditions of low speed, high load, and is characterized by a high COF while conversely hydrodynamic lubrication occurs under conditions of high speed and low load, and is characterized by a low COF. Using a DHR-2 Rheometer (TA Instruments), we applied a 5 N load and held it constant, while the radial velocity increased from 0.001 per rads to 50 per rads. Stribeck curves for all six mega HPGs formulations along with three controls, Pennzoil 80W-90 motor oil, Synvisc

One, and bovine synovial fluid (BSF), were constructed by plotting COF against the Hersey number (velocity × viscosity/load; Fig. 3). Synvisc One and BSF were selected as controls based on their different compositions (a lightly crosslinked linear polymer of hyaluronic acid of high viscosity versus a low viscous, natural lubricating solution) and relevance to the soft natural surface next investigated. Pennzoil was chosen in order to verify our method of Stribeck curve construction[39]. Additionally, Synvisc One and BSF lubricate via different mechanisms, BSF aids in mixed mode lubrication within joints, while Synvisc One is a fluid lubricant. The mega HPGs-1, -2, and -3, at both 7 and 23 w/v%, possess viscosities on the order of BSF and display boundary mode lubrication. The higher weight percent mega HPGs exhibit boundary mode lubrication at a consistently higher velocity with a molecular weight and concentration dependence. At 23 w/v%, the mega HPGs transition into mixed mode lubrication from boundary mode at increasing Hersey number with increasing molecular weight and with COFs equal to but speeds greater than BSF, despite their similarities in viscosities to BSF (Fig. 3a, b). Consequently, mega HPGs demonstrate Stribeck curves similar to the hyaluronic acid solution, Synvisc, except that the mega HPGs are 100× less viscous.

For the natural soft surface, we selected articular cartilage for evaluation of the mega HPGs as lubrication of this surface is key for bodily movement, protection from wear, and prevention of osteoarthritis[10,40–44]. Previously, linear[10,45] and brush polymers[40,46,47] have been explored for lubricating cartilage as these materials present controlled electrostatic interactions and hydration. Additionally, with the brush structures, the tilting and/ or the physical thinning of the polymer chains improves the lubricant properties[48,49]. Cartilage is a hydrated porous elasto-hydrodynamic material[12,13], where upon initial loading inter-stitial aqueous fluid supports most of the applied load by creating an interposed lubricating layer between the articulating surfaces. At creep equilibrium, most of the interstitial fluid has extruded, the deformed cartilage is softer and weeping lubrication is depleted. We conducted mechanical friction tests (BOSE Electroforce 3200) on harvested mated bovine osteochondral plug pairs (7 mm diameter, $N = 6$ pairs) in unconfined geometry (Fig. 4)[10]. After incubating the samples in the test groups (saline, healthy BSF, human osteoarthritic synovial fluid, and 1, 3, and 9 MDa mega HPGs at 7 and 23 w/v%), the opposing cartilage surfaces were pressed against each other (200 kPa creep load) for 3 h, while submerged in the test solutions. Then, while under

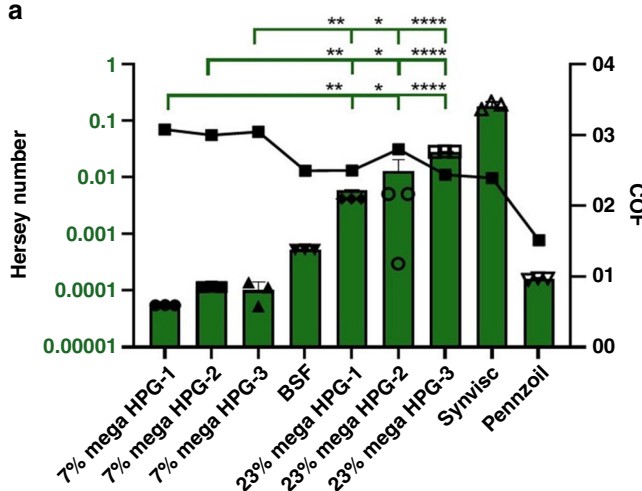

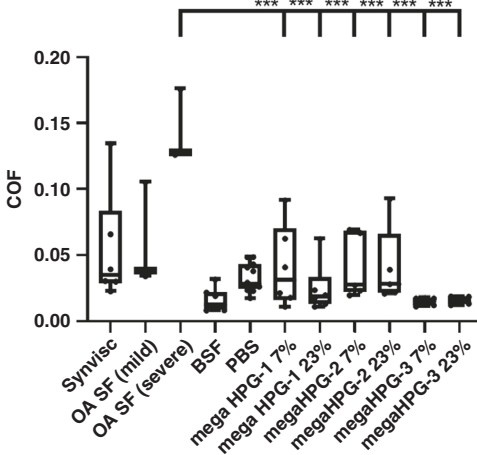

**Fig. 4 Determination of COF of mega HPGs.** COF values for cartilage on cartilage with each lubricant after equilibrating in creep, as shown with box-and-whiskers plot. Whiskers represent (min to max), bounds of box represent lower (25th percentile) and upper quartile (75th percentile), and center line represents median. Error bars represent standard deviation, $N = 3$ or greater replicates; one-way ANOVA used to compare groups, statistical differences indicated by asterisk, where *$p < 0.05$, **$p < 0.01$, ****$p < 0.0001$. For full list of statistical results see Supplementary Information.

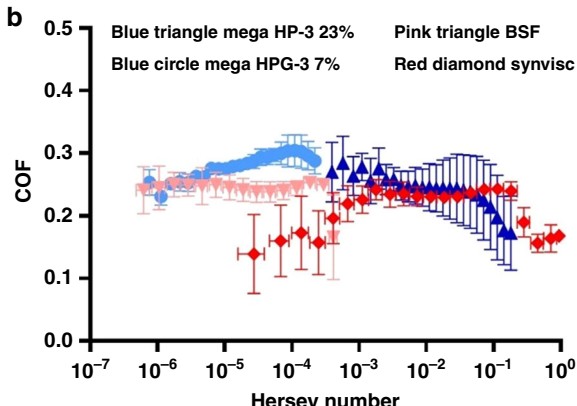

**Fig. 3 Lubrication characteristics of mega HPGs. a** Graph of the Hersey number at which each group transitions from boundary mode to mixed mode lubrication (left *y*-axis and green bars). The COF at the time each lubricant transitions from boundary to mixed mode lubrication (right *y*-axis, black symbols). Error bars represent standard deviation, $N = 3$ replicates; one-way ANOVA used to compare groups, statistical differences indicated by asterisk, where *$p < 0.05$, **$p < 0.01$, ****$p < 0.0001$. For full list of statistical results see Supplementary Information. **b**. Stribeck curves for best preforming mega HPG (mega HPG-3) at both 7 and 23 w/v%, as well as two controls, BSF and Synvisc One. Error bars represent standard deviation, $N = 3$ replicates.

load, rotation was applied for 120 s at an angular speed of 360°/s and the equilibrium COF was determined.

All of the mega HPGs lubricate the cartilage surface. The COF values for the mega HPGs are statistically equivalent to healthy BSF (positive control) and, importantly, significantly lower than the COF value obtained with human osteoarthritic synovial fluid (Fig. 4, $p < 0.0001$, one-way ANOVA; Supplementary Table 4). Between the different molecular weights of the mega HPGs, the 9 MDa mega HPG-3 exhibits a slightly lower COF, albeit not statistically significant. The performance of the mega HPG-3 is more consistent than the smaller mega HPG-1 and -2 lubricants, as evident by the lower standard deviation bars. With regards to the relative COF performance of the 9 MDa mega HPG-3 to other lubricants, it is challenging to compare as the extracted absolute values depends on the measurement geometry and protocol, as well as tissue type. Given the above caveats, the COF values of the mega HPGs are roughly an order of magnitude lower than the values reported for some bottle-brush copolymer lubricants

between cartilage and glass surfaces[47,50], while on par with linear polyelectrolyte polymers between cartilage and cartilage surfaces[51] (comparative study results presented in Supplementary Information COF chart; Supplementary Table 5). Analysis of the flow data reveals only the solution of mega HPG-1 (1 MDa) at 23% shear thins mimicking that of Synvisc, although with a smaller change in viscosity as a function of shear rate (Supplementary Fig. 12). We surmise that the mega HPG-2 and 3 (3 and 9 MDa polymers) are denser than the mega HPG-1. The hydrodynamic diameter of the mega HPG-3, for example, is only twice the size of the mega HPG-1 polymer, and therefore is less likely to entangle with itself. The mega HPG-1 extends into solution, shear thins, and is a non-Newtonian fluid. The mega HPG-3 maintains a constant viscosity across shear rate and acts as a Newtonian fluid. A Newtonian fluid lubricant is advantageous over a non-Newtonian one because the viscosity of the non-Newtonian lubricant reduces as a function of shear, displacing it from the surface and increasing the contact area between the surfaces. The mega HPGs exhibit viscosities similar to natural healthy synovial fluid, which are markedly lower than Synvisc (1226 mPas). Practically, this is advantageous as an 18 G needle is used to intra-articularly administer Synvisc, while mega HPGs easily flow through a 25 G needle. Finally, the cell compatibility of the mega HPGs against human chondrocytes and fibroblasts was evaluated at 48 h. The different molecular weight mega HPGs show close to 80% cell viability demonstrating high cytocompatibility of these ultra-high molecular weight dendritic polymers, with results similar to the saline control (Supplementary Fig. 13).

The mega HPGs acted as additive for enhanced liquid lubrication and join a class of tribological altering nanomaterials applied in lubrication engineering. The mechanism of anti-wear and friction reduction of these nano-lubricants include mending effect, colloidal effect, protective film, and third body material transfer[52,53]. The mega HPGs are 20–40 nm in diameter and in the size range of most metal nanoparticle additives, albeit they are softer materials and more compressible with a Young's Modulus of 7.9 kPa determined by atomic force microscopy (AFM)

measurements (Supplementary Fig. 14)[53]. No correlation is seen between COF and viscosity values suggesting that mega HPGs lubricate via a different mechanism than the extremely viscous Synvisc One. We propose that the mega HPGs, specifically the 3 and 9 MDa Newtonian fluid lubricants, function as interposed molecular ball bearings in water to reduce the COF between the stainless steel surfaces. We hypothesize a few modes of lubrication may be in effect during lubrication of cartilage with mega HPGs, particularly hydration shell lubrication described by Klein et al.[54–56], where our highly hydrated, water dense structures of the mega HPGs maintain a molecular water film at the cartilage surface, supporting heavy loads without being squeezed out while simultaneously rapidly relaxing[57]. Additionally, we suspect the 9 MDa mega HPG-3 is better retained on the tissue surface and does not get washed off, providing constant and lower COF compared to PBS, or other lubricants. This proposal is supported by experiments that show no shear thinning with the mega HPG-3 as it is a Newtonian lubricant, and follows the scenario in which Greene et al. suggests[58] that a mechanical trapping mechanism maintains a layer of immobilized HA between surfaces. This is similar to the 'ultra-filtration' hypothesis from Walker et al.[59], where water preferentially flows into the articular surface through the ~10 nm bovine cartilage pores[60], leaving larger molecules, such as the 20–40 nm diameter mega HPGs to aggregate at the leading edge of contact. How these proposed mechanisms relate to the mode of lubrication on cartilage—boundary, mixed, or hydrodynamic—requires further investigation.

In summary, we report the synthesis of ultra large dendritic polymers in the MDa range with high degrees of branching. The polymers are single molecule nanoscale objects with unique properties. The synthetic route affords control over the molecular weight and provides grams of material for study. The high water solubility, low intrinsic viscosity, compactness, nearly molecular weight independent intrinsic viscosity, hydration, and cell compatibility are important characteristics justifying the investigation of these polymers. The mega HPGs, we propose, are nanoparticulate lubricants acting as interposed ball bearings to reduce the COF between both hard and soft surfaces, and demonstrate rheological properties similar to a fluid lubricant. This unexpected result arises from the unique size and structure of the polymers. Size, structure, and composition dictate function and this is highlighted in the largest known protein, titin, at 3.7 MDa, which exhibits elastic properties and functions in muscle contraction[61]. Advances in synthetic polymer chemistry are providing routes to unique polymers and polymer architectures of unprecedented size and properties. These advances will propel our capability to prepare large single-entity molecular structures, as well as to conceive of self-assembled higher order complexes and to study the interactions of these materials with synthetic and natural substrates.

## Methods

**Materials**. All solvents and reagents were purchased from Sigma-Aldrich, Canada, unless otherwise mentioned. Glycidol was vacuum distilled over $CaH_2$ at 40–50 °C and stored over 4 Å molecular sieves. Anhydrous DMF and dioxane from Sigma-Aldrich were used without further purification. Deuterated solvent ($D_2O$, 99.8% D) was purchased from Cambridge Isotope Laboratories, Inc. Standard regenerated cellulose (RC) membranes (MWCO 50 and 10 kDa) were purchased from Spectrum, Inc., USA. NMR spectra ($^1$H, $^{13}$C, and $^{13}$C IG) were recorded on a Bruker Avance 300 and 400 MHz NMR spectrometers. Degree of branching of polymers was measured in deuterated water ($D_2O$) with a relaxation delay of 6 s, and it was calculated as per the reported procedure from the equation, DB = 2D/(2D + L), where D and L represent the intensities of the signals corresponding to the dendritic and linear units, respectively[62]. The absolute molecular weights of the polymers were determined by GPC on a Waters 2695 separation module fitted with a DAWN HELEOS II MALS detector coupled with Optilab T-rEX refractive index detector, both from Wyatt Technology, Inc., Santa Barbara, CA. GPC analysis was

performed using Waters ultrahydrogel columns (guard, linear and 120) and 0.1 N $NaNO_3$ buffer (pH = 7.0) was used a mobile phase, and dn/dc value for mega HPG used was 0.12 mL/g. The dn/dc values of the mega HPGs were determined independently; there was no molecular weight dependence. The hydrodynamic radii ($R_h$) of the polymers were obtained by quasielastic light scattering (QELS) detector using a Wyatt Internal QELS instrument (angle of measurement, 99.9°, laser $\lambda$ = 620 nm). The intrinsic viscosity measurements were performed on viscometer-II from Wyatt technologies in which 0.1 N $NaNO_3$ buffer (pH = 7.0) was used a mobile phase. Hydration of the polymers was determined using DSC (TA instruments, New Castle, DE, USA)[64]. Cryo-scanning electron microscopes size measurements were made on the FEI (Field Electron and Ion Company) Helios NanoLab 650 SEM with a focused ion beam (SEM-FIB) facility at 4D LABS, Simon Fraser University, Burnaby, Canada. Tc28a2 juvenile chondrocytes were purchased from EMD Millipore. MTT cell proliferation assay was purchased from ATCC. Absorbance readings for cell viability studies were measured at 570 nm on a SpectraMax 190 microplate reader from Molecular Devices.

**Synthesis of high molecular weight HPG macroinitiator[24]**. Trimethylolpropane (TMP; 120 mg, 0.894 mmol) was added to a flame-dried three-neck flask and dried under melting conditions (60 °C) and vacuum for 5 h. The dried TMP was partially deprotonated (33% of the total OH groups of TMP) with potassium methylate (25% MeOH, 70 μL, 0.237 mmol) under argon, and stirred for 30 min at room temperature (RT). Methanol was evaporated at 70 °C for 4 h under vacuum. The flask was connected to an overhead stirrer and dry dioxane (24 mL) was added followed by slow addition of glycidol (12 mL, flow rate of 0.5 mL/h) with a syringe pump under argon. After the addition of glycidol, the reaction mixture was stirred (rpm-150) for an additional 5 h. The reaction mixture was quenched with 0.5 mL of methanol after cooling it to RT. After decanting the supernatant (dioxane), the polymer was dissolved in 60 mL of methanol and precipitated from acetone (240 mL). The precipitation process was repeated twice (Caution: Make sure that a homogenous solution of the polymer in methanol was obtained before proceeding to precipitation with acetone. If it is taking long time for the dissolution of macroinitiator in methanol, the volume of methanol can be increased; however, same ratio of methanol and acetone should be maintained). The precipitated polymer was dissolved in deionized water (100 mL), neutralized by adding a few drops of 0.1 M HCl, and further purified by dialysis against water in a cellulose membrane (MWCO 10 kDa) for 2 days (with water replacement every 4–5 h). The polymer was freeze-dried and characterized by NMR ($^1$H and $^{13}$C NMR; Supplementary Figs. 1 and 2) and GPC-MALS analyses (conversion of monomer 100%, yield 70%; $M_w$ 840 kDa, Đ 1.2, Supplementary Fig. 3).

**Synthesis of mega HPG-1**. All the mega HPGs were synthesized by a modified macroinitiator approach[22]. The HPG macroinitiator ($M_w$ 840 kDa, Đ 1.2; 2.5 g, 0.034 mols of OH groups) was dissolved in MeOH (5.0 mL) in a flame-dried three-neck flask and a polymer film was made on the walls of the flask by slowly evaporating the solution under vacuum. The flask containing the polymer film was further dried under vacuum at 100 °C for overnight to completely remove traces of water and methanol. Thermogravimetric analysis was used to make sure that the polymer film was completely dried. The dried polymer was dissolved in dry DMF (35 mL) under Ar and KH suspension in oil (30%; 80 mg, 270 μL, 1 eq) was added. The flask was heated to 95 °C and stirred for 50 min to ensure that all the polymer was completely dissolved and turned into a yellow colored solution. The reaction flask was connected to an overhead stirrer under argon and the stirring speed was set at 150 rpm. To this homogenous solution, dried glycidol (8 mL) was added slowly (0.5 mL/h). After completion of glycidol addition, reaction was continued for ten more hours, then cooled it to RT. The solution was turned into a clear pale red colored solution. The conversion of the monomer was almost 100%, confirmed by $^1$H NMR spectroscopy. The polymer was dissolved in methanol (70–100 mL, make sure that no precipitate was found on the bottom of the flask) and precipitated in acetone (500 mL). The precipitation process was repeated two more times to remove the very small molecular weight fractions. The obtained precipitate was dissolved in water (100 mL), neutralized (pH 7) by dropwise addition of 0.1 M HCl, and was further purified by dialysis (RC dialysis membrane, MWCO 50,000 Da) against water for 5 days (water replacements every 8 h). The polymer was stored as aqueous solution at 4 °C (yield 80%). The experiment was repeated at least three times to determine the reproducibility (yield 81 ± 3.6%). The polymer was characterized by NMR, GPC-MALS, viscosity measurements, and quasielastic light scattering measurements (Table 1 and Supplementary Table 1). Caution: For the deprotonation of macroinitiator, the base, KH (in oil), is highly recommended. The KH solution should be homogenous before adding to the macroinitiator solution. Considering the pKa of the reagents involved in this deprotonation process, usage of either base KOMe or KH would be appropriate. However, if KOMe (25% in methanol) is used, the trace amounts of methanol can influence the polymer growth process resulting a bimodal distribution of HPG formed.

**Synthesis of mega HPG-2**. The HPG macroinitiator ($M_w$ 840 kDa, Đ 1.2; 2.5 g, 0.034 mols of OH groups) was dissolved in MeOH (5.0 mL) in a flame-dried three-neck flask and a polymer film was made on the walls of the flask by slowly evaporating the solution under vacuum. The flask containing the polymer film was

further dried under vacuum at 95 °C for 24 h to completely remove the traces of water and methanol. Thermogravimetric analysis was used to make sure that the polymer film was completely dried. The dried polymer was dissolved in dry DMF (35 mL) under Ar and KH (30% suspension in oil, 80 mg, 270 μL) was added. The flask was heated to 95 °C and stirred for 60 min to ensure that all the polymer was completely dissolved. The reaction flask was connected to an overhead stirrer under argon and the stirring speed was set up at 150 rpm. The solution color was turned to yellow. To this homogenous solution, dried glycidol (24 mL) was added slowly (1.4 mL/h). After completion of glycidol addition, reaction was continued for ten more hours, then cooled to RT. The color of the solution was turned into pale red. The conversion of the monomer was almost 100% confirmed by $^1$H NMR spectroscopy. Methanol (100 mL) was added to dissolve the polymer. A small amount of the precipitate was not soluble. 10 mL of DMF was added to completely to dissolve the polymer (make sure that no precipitate was found on the bottom of the flask). The polymer was precipitated from acetone (500 mL). The precipitation process was repeated two more times to remove the small molecular weight fractions. The obtained polymer was dissolved in water (100 mL), neutralized by dropwise addition of 0.1 M HCl, and was further purified by dialysis (RC dialysis membrane, MWCO 50,000 Da) against water for 5 days (water replacements every 8 h). The polymer was stored as aqueous solution at 4 °C (yield 85%). The experiment was repeated at least three times to determine the reproducibility (yield 78 ± 6.5%). The polymer was characterized by NMR, GPC-MALS, solubility, viscosity measurements, and QELS measurements (Table 1, Supplementary Tables 1 and 2).

**Synthesis of mega HPG-3.** The HPG macroinitiator ($M_w$ 840 kDa, Đ 1.2; 2.5 g, 0.034 mols of OH groups) was dissolved in MeOH (5.0 mL) in a flame-dried three-neck flask and a polymer film was made on the walls of the flask by slowly evaporating the solution under vacuum. The flask containing the polymer film was further dried under vacuum at 95 °C for 24 h to completely remove the traces of water and methanol. Thermogravimetric analysis was used to make sure that the polymer film was completely dried. The dried polymer was dissolved in dry DMF (35 mL) under Ar and KH suspension in oil (30%; ~80 mg, 270 μL) was added. The flask was heated to 95 °C and stirred for 30 min to ensure that all the polymer was completely dissolved. The reaction flask was connected to an overhead stirrer under argon and the stirring speed was set up at 200 rpm. To this homogenous solution, dried glycidol (50 mL) was added slowly (1.4 mL/h). After completion of glycidol addition, reaction was continued for ten more hours, then cooled it to RT. The conversion of the monomer was almost 100% confirmed by $^1$H NMR spectroscopy. Methanol (100 mL) was added to dissolve the polymer. A small amount of the precipitate was not soluble. A total of 10–20 mL of DMF was added to completely to dissolve the polymer (make sure that no precipitate was found on the bottom of the flask). The polymer was precipitated from acetone (500 mL). The precipitation process was repeated two more times, dissolved in water (100 mL), neutralized (pH 7) by dropwise addition of 0.1 M HCl, and was further purified by dialysis (RC dialysis membrane, MWCO 50,000 Da) against water for 5 days (water replacements every 8 h).The polymer was stored as aqueous solution at 4 °C (yield 74 %). The experiment was repeated at least three times to determine the reproducibility of the synthesis of mega HPG-3 (yield 74 ± 0.21%). The polymer was characterized by NMR, GPC-MALS, solubility, viscosity measurements, and QELS measurements (Table 1, Supplementary Table 1).

**Solubility measurements.** The solubility of mega HPGs in water was measured, and compared with other linear polymers (PEO and PVA purchased from Sigma-Aldrich, ON). The mega HPGs were initially dissolved in water (100 mg in 1 mL) using vortex mixing. Additional amount of mega HPGs were added until the solution reached saturation (Supplementary Table 1). The solution was equilibrated overnight to measure consistency. The similar protocol was repeated for the other linear polymers until the solution become a stable gel.

**Determination of hydration of mega HPG.** Hydration of the polymers was determined using DSC[65]. The mega HPG solution was prepared in water (10% W/W). A total of 20 μL of the solution was loaded into a Tzero aluminum hermetic sample pan and closed with appropriate lids. The sample pan was cooled down to −20 °C and warmed to −5 °C at the rate of 2 °C/min. Heating of the sample was further continued from −5 °C to +5 °C at the rate of 0.2 °C/min and to +20 °C at the rate of 2 °C/min. The enthalpy of fusion of polymer solution or pure water was determined by integrating the area under the respective peak on DSC trace (Supplementary Table 2). An empty pan was used as a reference.

The number of water molecules bound per polymer was calculated using the following equation.

$$[N_{bw}] = [\Delta H_{fo}(Wt_w − Wt_p) − \Delta H_{fps} \times Wt_{ps}]/[\Delta H_{fo} \times MW_{H2O} \times Np]$$

$N_{bw}$ = number of water molecules bound per polymer,
$\Delta H_{fps}$, $\Delta H_{fo}$ = fusion enthalpy of polymer solution and pure water, respectively,
$Wt_w$, $Wt_p$, $Wt_{ps}$ = weight of the pure water, pure polymer alone, and polymer solution, respectively.
$Np$ = number of moles of polymer taken.

**Cryo-SEM measurements.** The morphology of the mega HPGs were visualized using cryo-SEM assembled with a focused ion beam SEM-FIB equipped with Quorum PP3010T cryo sample preparation system. A small volume of sample (10 μL) solution was loaded onto the sample carrier and immersed in slush liquid nitrogen for rapid freezing. The carrier with the frozen solution pellet is quickly vacuum transferred to the sample preparation stage cooled at −140 °C for fracturing the pellet. The fractured pellet was further transferred to the SEM stage cooled also at −140 °C for imaging the fractured surface, with an electron beam of 1 keV and 13 pA. The optimal concentration of the mega HPGs in Millipore water was 0.1 mg/mL. The average size of the mega HPGs were calculated from the ten images with ~500 particles.

**Cell viability measurements.** Cell viability was assessed in Tc28a2 juvenile human chondrocytes and 3T3 murine fibroblast cells using the (3-(4,5-dimethylthiazol-2-yl)-2,5-diphenyltetrazolium bromide) tetrazolium reduction (MTT) assay as per previously reported protocol[10]. Cells were seeded at 10,000 cells/well and allowed to settle and adhere for 24 h before addition of mega HPG or control samples. Mega HPG samples (10 μL) were prepared in regular growth media (DMEM) for (90 μL) to obtain a final polymer concentration of 1.25 mg/mL. Cells were incubated with the mega HPG samples for 48 h at 37 °C. Wells containing 50% DMSO was used as a positive control. Wells containing saline and media in similar volumes to the polymer samples were used as normal controls. After incubation, cells were washed with PBS three times followed by the addition of 100 μL of fresh media and 10 μL of 12 mM MTT reagent (from ATCC). After 4 h, 100 μL of sodium dodecyl sulfate-HCl was added to solubilize the generated formazan. After solubilization for 2–4 h, absorbance at 570 nm was read on a SpectraMax 190 microplate reader (Molecular Devices) and compared to saline controls. Three technical replicates were conducted per sample, and each study was independently repeated in triplicate. Average values and standard deviation are reported.

**Lubrication measurements.** Viscosity measurements were performed on a TA Instrument AR2000 rheometer using a 2° aluminum cone and a 47 μm gap at 25 °C. Viscosity was then averaged across shear rates from 10 to 100 per s. Stribeck curves were constructed on a DHR-2 rheometer (TA instruments) with a stainless steel ring on plate tribology geometry attachment. A total of 300 μL of each lubricant were applied to the stainless steel surface and a 5 N load was applied, radial velocity increased from 0.001 rad/s to 50 rad/s while the load was held constant. This test was replicated three times for each lubricant, using fresh lubricant for all measurements. Average COF values and Hersey numbers, along with standard deviation and Stribeck curves for all formulations of mega HPG along with Synvisc One and BSF as controls. The values of 7 and 23% for mega HPGs were chosen as to ensure a 3× difference in amounts to better observe potential differences. Cartilage on cartilage COF measurements, using 7 mm diameter bovine osteochondral plugs, were performed on a Bose Electroforce 3200. Cartilage plugs were cored from skeletally mature bovine knees using a diamond tipped drill bit. Plugs were incubated in 0.5 mL of lubricant at RT overnight prior to testing. An 8 N (200 kPa) creep load was applied to paired osteochondral plugs. After 3 h of compression to equilibrate the tissue in creep, equilibrium friction measurement was made. Friction measurements used an angular velocity of 360°/s (effective velocity = 14.7 mm/s) for 120 s. COF ($\mu$) was calculated from $\mu = (3/2) \times (\tau/Nr)$, where: $\tau$ = torque, $N$ = load, $r$ = plug radius[62]. Sampling frequency was 10 Hz.

**AFM measurements.** AFM was performed using an MFP-3D microscope (Asylum Research; Santa Barbara, CA). The contact mode was applied using a silicon nitride tip with a nominal spring constant of 40 pN/nm. Mega HPG-3 was chemically adhered to the surface of epoxide functionalized glass slides. A total of 5 mg of mega HPG-3 was diluted in 50 mL of dimethylformamide. Stoichiometric amount of sodium hydride was then added. Ten microliters of this solution at concentration of (0.1 mg/mL) was dropped on the surface of the epoxide glass slide and kept at 60 °C overnight. The glass surface was rehydrated in PBS just before experimentation. The tip was slowly lowered to the surface to confirm contact without damage. Tip calibration was done on the bare glass surface, which acts as an infinitely hard surface, to set both a baseline deflection and virtual deflection correction. This determines the cantilever's inverse optical lever sensitivity (InvOLS; unit: m/V). The spring constant was verified via thermal tuning and was always within the range dictated by Bruker (within 30 pN/nm). After calibration, an area scan was performed in tapping mode prior to indentation to confirm the presence and density of the polymer. Clumps of polymer were chosen for indentation based on shape and size. Force spectroscopy was then obtained over a 2 μm extension length and a 1000 nm/s approaching and retreating velocity. The tip was triggered at a set-point of 0.75 V of deflection. Once an indentation was performed, the raw distance of the tip along the z-direction was converted into an indentation depth using the InvOLS.

## Data availability

The authors declare that all the data supporting these findings of this study are available within the paper and its supplementary information file.

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

## Acknowledgements

The authors thank the Macromolecular Hub (CBR) and 4D LABS facility (Simon Fraser University), for the use of their research facilities. We thank Dr. Catalina Bordeianu for her input. The authors acknowledge the funding by Canadian Institutes of Health Research (CIHR), Natural Sciences and Engineering Council of Canada (NSERC), Canada Foundation for Innovation (CFI), The BUnano Innovation Center, and the National institutes of Health (NIH; R01AR066621). J.N.K. holds a Career Investigator Scholar award from the Michael Smith Foundation for Health Research (MSFHR). S.A. acknowledges a MSFHR postdoctoral fellowship. L.T. acknowledges funding from NSERC CGS-M and the NSERC CREATE NanoMat Program. T.B.L. acknowledges funding from the NIH (F31 AR075386). J.T.A.M. acknowledges funding from Finnish Cultural Foundation, Päivikki and Sakari Sohlberg Foundation, and Orion Research Foundation.

## Author contributions

P.A. and S.A. synthesized and characterized mega HPGs. T.B.L., J.T.A.M., and R.C.S. performed lubrication studies of mega HPGs. B.D.S. assisted in analyzing lubrication data. L.E.T. performed cellular toxicity studies. Manuscript was written by S.A., J.T.A.M., M.W.G., and J.N.K. with the help of other authors. M.W.G. and J.N.K. provided the grant support and supervision of the project.

## Competing interests

The authors declare no competing interests.
