## [Peer Review File · Nature Communications]

Reviewers' comments:

Reviewer #1 (Remarks to the Author):

This is a well-crafted manuscript that describes the large scale synthesis of a glycerol dendrimer, and the resulting structure's capacity to lubricate two different surface types, using two different methods. The report is interesting and has merit, but there are a few concerns that need to be addressed (below).

1. The conditions under which the cartilage plug studies are conducted can not be accurately defined as boundary mode. The rotation method that is used here is not characterized with respect to the different lubrication regimes (as was done with the Stribeck analysis against stainless steel). Therefore the premise that these mega-macromolecules lubricates cartilage under boundary conditions is not supported by this study.
2. Figure 3: It is strange that the statistical comparisons are made between SF from severe OA joints. Also, the PBS COF is on par with the macromolecule lubrication results (lower even), which is very odd. This figure really does not support the premise that these structures are effective lubricants.
3. The structure in Figure 1A of the final dendrimer does not reflect the degree of branching that is reported in Table 1. If I read this correctly, Figure 1A would depict a 1.0 degree of branching, whereas Table 1 states the degree of branching is ~ 0.5 .
4. All NMR spectra should be annotated with the chemical structures.

Reviewer #2 (Remarks to the Author):

The authors reported the preparation and characterization of mega hyperbranched polyglycerols (mega HPGs) with molecular weight in the millions Dalton range. Several unique properties such as high water solubility, low intrinsic viscosity and compactness enable this nanometer-scale mega HPG could function as boundary lubricant to reduce the coefficient of friction (COF) between both hard and soft surfaces. However, their synthetic strategy is not new and modified from the previously reported macro-initiator approach, the authors did not provide clear procedures nor they dig into the reaction to disclose the real reason why they could synthesize mega HPGs with such high molecular weight, which is fairly important for understanding the synthesis of this type materials. Therefore, I think it is not suitable for publication in Nature Communications. Besides that, the authors are suggested to address the following issues before they submitted to the other journals.

- 1) The monomer structure shown in Fig. 1A is incorrect and there are also some mistakes in the supplementary information.
- 2) The authors are suggested to provide clear description of their method (i.e. the amounts of micro-initiators. Ratios, monomer conversion, why bases utilized for the synthesis of micro-initiators and mega HPGs is different?) and make comparison with the previous macro-initiator approach to show their improvement and advantages.
- 3) There is no description on the preparation of 7% and 23% mega HPGs and why these two were chosen for the lubrication measurement.
- 4) As a potential material for lubricant, the authors should also provide the suitable application range of viscosity that could be used as synovial fluid.
- 5) 1h is too short for the evaluation of the cell compatibility, it is suggested to perform the cytotoxicity studies of Mega HPGs for at least 24 h.

6) The authors mentioned brush or bottle type linear polymers with controlled electrostatic interactions and hydration can be used as boundary lubricants, but there is no comparison of their performance with mega HPGs.

Reviewer #3 (Remarks to the Author):

The paper by Anilkumar et al. describes the synthesis and the characterization of hyperbranched polyglycerols that exhibit a molecular weight up to nearly 10 MDa. The authors can clearly prove that they reached this high molecular weight indeed. Moreover, they can demonstrate that aqueous solutions of these mega-HPGs exhibit low friction and may have interesting medical applications. I believe that this report possesses the sufficient novelty to warrant publication as a communication to Nature. However, there are still some points which should be improved prior to publication:

1. I believe that Figure 12B is very instructive and should be moved to the main text. It should be shown in a double-logarithmic axis. I understand that the intrinsic viscosity should be nearly independent of the molecular weight. However, its value starts to increase considerably for molecular weights beyond 1 MDa. Why is this so? Do the authors have an explanation for this observation?

2. My main problem with the paper is related to the description of the lubrication properties of the aqueous solutions. To my understanding, the excellent lubrication of these solutions is simply due to the low intrinsic viscosity in combination of a strong dependence on shear rate. Is this so? I guess that most of the ordinary readers of this journal are not familiar with Hersey numbers and Stribeck curves. Therefore, I suggest the authors show ordinary flow curves in which the shear viscosity is plotted against the shear rate. This information should already be available from the measurements done so far and would be far more instructive than the present figure 2. What is the physical reason for the low friction? The authors may find it revealing to study the respective investigations of J. Klein and coworkers who analyzed the friction between polymeric surfaces in detail. In turn, Figure 3 could be easily moved into the SI. I find the discussion of the various data gathered in this figure not very instructive and near to incomprehensible.

3. Figure 11A: The cryo-TEM looks as if the systems would be hollow spheres. Could the authors comment on this observation?

Responses Reviewer's Comments (Manuscript ID: NCOMMS-19-32024-T)

Reviewer #1 (Remarks to the Author):

Comment. This is a well-crafted manuscript that describes the large-scale synthesis of a glycerol dendrimer, and the resulting structure's capacity to lubricate two different surface types, using two different methods. The report is interesting and has merit, but there are a few concerns that need to be addressed (below).

Response. We thank the reviewer for the positive and helpful comments to improve the manuscript.

Comment 1. The conditions under which the cartilage plug studies are conducted cannot be accurately defined as boundary mode. The rotation method that is used here is not characterized with respect to the different lubrication regimes (as was done with the Stribeck analysis against stainless steel). Therefore, the premise that these mega-macromolecules lubricates cartilage under boundary conditions is not supported by this study.

Response 1. The reviewer is correct, and we appreciate the careful analysis. The lubrication mode (boundary, mixed, (elasto)hydrodynamic) is dependent on the articular cartilage pressurization [DOI: 10.1016/j.jbiomech.2009.04.040 (*J Biomech.* **2009**, *42*, 1163)]. We equilibrated our samples to minimize this pressurization effect and called it boundary mode as the surfaces would not be repelled due to the exuding interstitial liquid. Our chosen terminology is confusing/incorrect, and we apologize. We did not vary the velocity, as in the stainless-steel experiments. Prolonged series, i.e., preservation and repetitive loading, to create Stribeck curves (each measurement 4 h, and swell period in between) leads to degradation and loss of cartilage proteoglycans, altering the biomechanics [DOI:10.1115/1.4000991 (*J. Biomech. Eng.* **2010**, *132*, 064502), DOI:10.1016/j.jmbbm.2016.09.040 (*J. Mech. Behav. Biomed. Mater.* **2017**, *65*, 734), DOI:10.1002/jor.1100020109 (*J. Orthop. Res.* **1984**, *2*, 55), DOI:10.1002/jor.23782 (*J. Orthop. Res.* **2018**, *36*, 1456)]. Additionally, we used a velocity of 15 mm/s while earlier studies of cartilage boundary lubrication used <1 mm/s [DOI:10.1016/j.jbiomech.2008.03.043 (*J. Biomech.* **2008**, *41*, 1910), DOI:10.1016/j.joca.2006.06.005 (*Osteoarthr. Cartil.* **2007**, *15*, 35)] as the higher velocities are more physiologically like. Burris et al. [DOI: <https://doi.org/10.1007/s11249-019-1158-7> (*Tribol. Lett.* **2019**, *67*, 46)] showed that it takes sliding speeds ~ 20 mm/s to affect cartilage compression and friction and transition out of

Figure 1. How Sliding and Hydrodynamics Contribute to Articular Cartilage Fluid and Lubrication Recovery. (Figure 6, from Burris et al., *Tribol. Lett.* **2019**, *67*, 46).

boundary lubrication (Figure 1 below. They used identical pressure ~200kPa.). In our experimental setup, the lubrication mode is likely mixed in the presence of the lubricant test group and the higher load, when cartilage is equilibrated.

We have edited the text to state this limitation, included the above references, added flow curves to the SI (Supplementary Fig. 12), cited more literature for comparison, and softened our conclusion regarding the lubrication mechanism of these interesting and very new single molecule lubricants: Discussion, Page 6 (line 33)-7(line 2). “We propose that the mega HPGs, specifically the 3 and 9 MDa Newtonian fluid lubricants, function as interposed molecular ball-bearings in water to reduce the COF between the stainless-steel surfaces. We hypothesize a few modes of lubrication may be in effect during lubrication of cartilage with mega HPGs, particularly hydration shell lubrication described by J. Klein et al., [DOI: 10.1038/s41467-017-01421-7 (Nat. Commun. 2017, 8, 1546), DOI:10.1038/nature05196 (Nature 2006, 444, 191) , DOI:10.1038/nature01970 (Nature 2003, 425, 163)], where our highly hydrated, water dense structures of the mega HPGs maintain a molecular water film at the cartilage surface, supporting heavy loads without being squeezed out while simultaneously rapidly relaxing [DOI: 10.1021/nn5062707 (ACS Nano 2015, 9, 2614)]. Additionally, we suspect the 9 million Dalton mega HPG-3 is better retained on the tissue surface and does not get washed off, providing constant and lower COF compared to PBS, or other lubricants. This proposal is supported by experiments that show no shear thinning with the mega HPG-3 as it is a Newtonian lubricant and follows the scenario in which Greene suggests [DOI:10.1073/pnas.1101002108 (PNAS 2011, 1),] that a "mechanical trapping" mechanism maintains a layer of immobilized HA between surfaces. This is similar to the ‘ultra-filtration’ hypothesis from Walker et al. [DOI: 10.1136/ard.27.6.512 (Ann. rheum. Dis. 1968, 27, 512)], where water preferentially flows into the articular surface through the ~10nm bovine cartilage pores [DOI: 10.1016/j.micromeso.2017.01.005 (Microporous and Mesoporous Mater. 2017, 241, 238)], leaving larger molecules, such as the 20 – 40 nm diameter mega HPGs to aggregate at the leading edge of contact. How these proposed mechanisms relate to the mode of lubrication on cartilage – boundary, mixed, or hydrodynamic – requires further investigation.”

Comment 2. Figure 3: It is strange that the statistical comparisons are made between SF from severe OA joints. Also, the PBS COF is on par with the macromolecule lubrication results (lower even), which is very odd. This figure really does not support the premise that these structures are effective lubricants.

Response 2. The statistical comparisons were performed across all the samples and the complete analysis is found in Supplementary Information (Supplementary Table 3-4). We chose to compare the mega-macromolecules lubricants of three different sizes and two different concentrations to several control lubricants including bovine synovial fluid (healthy normal positive control), human OA synovial fluid (negative and clinically relevant control), Synvisc (commercial OA treatment), and PBS. We have changed Figure 4 from bar plot to box plot to better display the data and the groups. The reviewer is correct that the results for the PBS are on par with the smaller mega-macromolecules lubricants. We hypothesize that this may be a combination of native cartilage’s surface attached macromolecules and the non-shear thinning properties of the smaller mega-HPGs, and/or hydration shell lubrication described by Klein et al., [DOI: 10.1038/s41467-017-01421-7 (Nat. Commun. 2017, 8, 1546), DOI:10.1038/nature05196 (Nature 2006, 444, 191), DOI:10.1038/nature01970 (Nature 2003, 425, 163)]. Surface-attached hydration layers occur as a result of water molecules bound to ionized surfaces in aqueous electrolyte solutions. Cartilage being an anionic material, when exposed to a bath solution of electrolytes, ions will condense over all ionizable sites, attracting water to the surface, creating charge bound water molecules on the surface of the cartilage. These charge bound water molecules create hydration layers, and as J. Klein suggests, when these two hydration layers overlap, water moves relatively freely in the overlapping region, thereby effectively acting as a low viscosity lubricant. Additionally, we hypothesize that apart from hydration shell lubrication, the lubrication may work in combination with surface-attached hydrogel layers [DOI: <https://doi.org/10.1016/j.triboint.2019.02.045> (Tribol. Inter. 2019, ASAP)].

Our aim was to study the native cartilage lubricating properties of these mega-macromolecules as a function of size. The lubricating property of the 9 million Dalton mega HPG is slightly better than PBS and outperforms OA synovial fluid and Synvisc. We suspect the 9 million Dalton mega HPG-3 is better retained on the tissue surface providing a highly hydrated state and does not get washed off, providing constant and low COF compared to the PBS, or other lubricants. This scenario is aligned with which Greene proposes [DOI:10.1073/pnas.1101002108 (PNAS 2011, 1)] that a "mechanical trapping" mechanism maintains a layer of immobilized hydrated HA between surfaces. Similarly, the 'ultra-filtration' hypothesis from Walker et al., [DOI: <http://dx.doi.org/10.1136/ard.27.6.512> (Ann. rheum. Dis. 1968, 27, 512)], suggests that water preferentially flows into the articular surface through the ~10 nm bovine cartilage pores leaving larger molecules, such as mega HPGs with diameters of 20 – 40 nm, to aggregate at the leading edge of contact [DOI: 10.1016/j.micromeso.2017.01.005 (Microporous and Mesoporous Mater. 2017, 241, 238)] (page-6, line 33). The results are interesting, and these first studies support continued investigation into the mechanism of lubrication on soft tissues.

We have updated the Figure 4, edited the main text and SI and included the above information to improve the clarity of the text. Please see our previous action.

Comment 3. The structure in Figure 1A of the final dendrimer does not reflect the degree of branching that is reported in Table 1. If I read this correctly, Figure 1A would depict a 1.0 degree of branching, whereas Table 1 states the degree of branching is ~0.5.

Response 3. We provided a schematic representation of mega HPG structure in figure 1A, with a degree of branching approximately 50%. However, in the revised manuscript, we have modified the scheme and redrew the structures of macroinitiator as well as mega HPGs with approximate degree of branching of 56-57% (highest degree of branching noticed with these systems). We used the following equation to calculate the degree of branching of hyperbranched polymers derived from AB₂ type monomers through anionic ring opening polymerization [DOI: 10.1021/ma990090w (Macromolecules 1999, 32, 4240–4246)].

$$\text{Degree of branching} = 2D/(2D+L_{13}+L_{14})$$

The relative abundance of dendritic (D), linear (L₁₃), and liner (L₁₄) linkages of the dendritic/hyperbranched structures was determined by ¹³C inverse-gated NMR spectroscopy. Based on these data, we redrew the structures of mega HPG based on the abundance of the dendritic (D- 27 ± 0.71%), linear (L₁₃- 14 ± 1.2%), linear (L₁₄- 27 ± 0.07%), and terminal units (30 ± 2.0%).

Figure 2. A schematic representation of synthesis of *mega* HPGs. The color code show different linkages present within the *mega* HPGs.

Comment 4. All NMR spectra should be annotated with the chemical structures.

Response 4: We agree and apologize for overlooking this. We did not provide the structural annotation because the NMR structural analysis for small molecular weight HPGs was previously published. No appreciable differences were shown by *mega* HPGs in comparison to their small molecular weight counterparts in terms of NMR characterization. In the revised manuscript, all the NMR spectra are included with appropriate annotation. The $-CH-$ and $-CH_2-$ protons of the *mega*-HPG appear around 3.0-4.4 ppm on the 1H NMR spectrum (Fig. 1b and Supplementary Fig. 1, 6, and 7). We used ^{13}C inverse-gated NMR spectroscopy in order to distinguish the dendritic, linear, and terminal carbon units of the *mega* HPGs. All the peaks in the ^{13}C NMR spectra in both main text (Fig. 1c) and supplementary information (Supplementary Fig. 2, 8, and 9) are now annotated. The structural analysis of *mega* HPG-3 is shown below.

Figure 3. NMR structural characterization of *mega* HPG-3. 1H NMR (left) and ^{13}C NMR spectra of the molecule.

Reviewer #2 (Remarks to the Author):

Comment. The authors reported the preparation and characterization of mega hyperbranched polyglycerols (mega HPGs) with molecular weight in the millions Dalton range. Several unique properties such as high-water solubility, low intrinsic viscosity and compactness enable this nanometer-scale mega HPG could function as boundary lubricant to reduce the coefficient of friction (COF) between both hard and soft surfaces. However, their synthetic strategy is not new and modified from the previously reported macro-initiator approach, the authors did not provide clear procedures nor they dig into the reaction to disclose the real reason why they could synthesize mega HPGs with such high molecular weight, which is fairly important for understanding the synthesis of this type materials. Therefore, I think it is not suitable for publication in Nature Communications. Besides that, the authors are suggested to address the following issues before they submitted to the other journals.

Response. We thank the reviewer for noting the unique properties such as high-water solubility, low intrinsic viscosity and compactness as well as potential of these new molecules as single molecule lubricants. Additionally, this is the first time such large globular semi-dendritic polymers have been prepared in a single pot. We agree with the reviewer that we did not provide sufficient details in the experimental section to explain why this synthetic strategy works and how we were able to synthesize mega HPGs with such high molecular weight. We apologize for this oversight and made every effort to address this concern in the revised manuscript. We have added additional text to clearly state the advantages of this new/improved method as well as addressed the other comments below. These changes have further strengthened the manuscript, and we strongly believe its appropriateness for Nature Communications.

We performed a major revision by giving significant additional synthetic details of mega HPG (Supplementary Information: Synthesis of high molecular weight HPG macro-initiator and Synthesis of mega HPGs).

Further, we highlighted the importance in the synthesis of mega HPGs (please see comment 2/response 2 below) and also the novelty of the observation that mega HPGs function as single molecule lubricants (please see comment 4/response 4) demonstrating that these molecules exhibited lubrication properties like nanoparticle lubricants.

Comment 1. The monomer structure shown in Fig. 1A is incorrect and there are also some mistakes in the supplementary information.

Response 1. We apologize and thank you for catching the typo. We have addressed the concern and the new scheme is provided in the revised manuscript (Fig. 1a). Figures in the Supplementary information have also been modified.

Figure 4. A schematic representation of synthesis of mega HPGs. The color code shows the linkages within mega HPGs.

Comment 2. The authors are suggested to provide clear description of their method (i.e. the amounts of micro-initiators. Ratios, monomer conversion, why bases utilized for the synthesis of micro-initiators and mega HPGs is different?) and make comparison with the previous macro-initiator approach to show their improvement and advantages.

Response 2. We thank the reviewer to bring up these points. The supporting information in the revised manuscript has been extensively modified to include the following additional details (Supplementary Information: Synthesis of high molecular weight HPG macro-initiator and Synthesis of mega HPGs). The previous macroinitiator approach reported in neat conditions only generated HPGs up to 20 kDa [DOI: 10.1021/ma802701g (Macromolecules 2009, 42, 3230–3236)]. In our current manuscript, we synthesized polymers up to 10 million Dalton via a combination of macroinitiator approach and solvent based polymerization approach in a single pot. This is a significant achievement considering the difficulty in synthesizing high molecular weight HPGs. Importantly, we overcome a major hurdle in generating ultra-high molecular weight semi dendritic HPGs via anionic ring opening polymerization using combined macroinitiator and solvent based approach.

Specifically, we have added more details on the synthesis of macroinitiator (which itself is 840 kDa) and mega HPGs, including monomer conversion, amount of initiator, and precipitation conditions, etc. (See Supporting Information, pages 3-5). For the deprotection of macroinitiator, the base, KH (in oil), is highly recommended. Considering the pKa of the reagents involved in this deprotonation process, use of either base (KOMe or KH) would be appropriate. However, the use of KOMe (25% in methanol) will result in trace amounts of methanol in the polymerization medium which can influence the polymer propagation process leading to bimodal distribution of polymers. Thus, KH is more appropriate. In addition, the KH solution should be homogenous before adding to the macroinitiator solution. The monomer conversion is quantitative; however, the yields were varied based on the propagation of the polymer chains and purification methods. We have repeated these experiments at least 3 times to verify the repeatability. We have added this information to the revised manuscript and supplementary information (see Supporting Information, page 4).

Further, we note that the synthesis of dendrimers of similar molecular weight has not been achieved previously. In addition to its application as single molecule lubricants, the new mega macromolecules could offer its use in diverse fields including conjugation of small molecular drugs, ligands, antibiotics, proteins, peptides, radiopharmaceuticals, and imaging tools with highly improved activity compared to linear polymers owing to their low intrinsic viscosity, high biocompatibility and presence of large number of functional groups. We anticipate that these new mega macromolecules will have extensive material and biomedical applications.

Comment 3. There is no description on the preparation of 7% and 23% mega HPGs and why these two were chosen for the lubrication measurement.

Response 3. We have selected two concentrations in order to determine if there is a concentration dependence on lubrication. The values of 7 and 23% were chosen as to ensure a 3X difference in amounts to better observe potential differences. This information is included in the revised supplementary information (page 6).

Comment 4. As a potential material for lubricant, the authors should also provide the suitable application range of viscosity that could be used as synovial fluid.

Response 4. The mega HPGs exhibit viscosities similar to natural healthy synovial fluid (<100 mPas), which are markedly lower than commercial Synvisc (>1000 mPas). Additionally, this low viscosity is advantageous as the mega HPGs easily flow through a 25 G needle and do not require a 18G needle for intra-articular administration as with Synvisc. A suitable application range of viscosity would be 1-10 Pa*s. We have added flow experimental data and charts to the SI (Supplementary Fig. 12) so that the readers know the viscosity properties. This was suggested by Reviewer #3, and we appreciate the comment. The viscosity is not dependent on the shear rate for the 7 wt% 1, 3, and 9 MDa polymer solutions, and these compositions behave as Newtonian fluids. Whereas the solution of 1 MDa at 23% demonstrates shear thinning behavior mimicking that of Synvisc although with a smaller change in viscosity as a function of shear rate. The 23 wt% solution of 3 and 9 MDa polymers behave as Newtonian fluids. We surmise that the 3 and 9 MDa polymers are denser, and more compact than the 1MDa polymer and do not entangle with each other, as the hydrodynamic diameter of 9 MDa polymer is only twice the size of the 1 MDa polymer. The 1 MDa polymer extends into solution, shear thins, and is a non-Newtonian fluid. The 9 MDa polymer maintains a constant viscosity across shear rate. If the viscosity did reduce as a function of shear, the lubricant would be displaced, and surfaces would come into contact and thus a Newtonian fluid lubricant is advantageous over a non-Newtonian one.

We have added the following text to the revised manuscript: "Analysis of the flow data (Supplementary Fig. 12) reveals only the solution of 1 MDa at 23% shear thins mimicking that of Synvisc, although with a smaller change in viscosity as a function of shear rate. We surmise that the 3 and 9 MDa polymers are denser than the 1MDa polymer. The hydrodynamic diameter of the 9 MDa polymer, for example, is only twice the size of the 1 MDa polymer, and therefore is less likely to entangle with itself. The 1 MDa polymer extends into solution, shear thins, and is a non-Newtonian fluid. The 9 MDa polymer maintains a constant viscosity across shear rate and acts as a Newtonian fluid. A Newtonian fluid lubricant is advantageous over a non-Newtonian one because the viscosity of the non-Newtonian lubricant reduces as a function of shear, displacing it from the surface and increasing the contact area between the surfaces (page 6, Paragraph-2)."

Comment 5. 1h is too short for the evaluation of the cell compatibility, it is suggested to perform the cytotoxicity studies of Mega HPGs for at least 24 h.

Response 5. We thank the reviewers for this comment. We have performed additional experiments to support our finding regarding the cell compatibility. Cell viability of *mega* HPGs-1, 2, and 3 towards (a) Tc28a2 juvenile human chondrocytes, and (b) 3T3 murine fibroblast cells was assessed after 48 h of exposure. Cells were incubated with either *mega* HPGs, saline, or DMSO for 48 h at 37 °C. After washings, the metabolic activity of the cells was assessed by MTT assay. High cell viability of the *mega* HPGs ($\geq 80\%$) toward all cell lines used validated the high compatibility of the *mega* HPGs. Please see the figure below, which has been included into the revised SI (see Supplementary Fig. 13).

Figure 5. Cell viability of *mega* HPGs-1, 2, and 3 (1.25 mg/ml) towards Tc28a2 juvenile human chondrocytes (a) and 3T3 murine fibroblast cells (b). Cells were incubated with either *mega* HPGs, saline, or DMSO for 48 h at 37 °C. After washings, the metabolic activity of the cells was assessed by MTT assay. Six replicates were performed and each study was repeated in quadruplicates. Average values and standard deviation are reported. Cell viability of the *mega* HPGs ($\geq 80\%$) irrespective of the cell line confirmed the high cell compatibility of the *mega* HPGs.

Comment 6. The authors mentioned brush or bottle type linear polymers with controlled electrostatic interactions and hydration can be used as boundary lubricants, but there is no comparison of their performance with *mega* HPGs.

Response 6. We thank the reviewer for the comment. In the beginning of the lubrication section (page 4, main text) we describe, and reference previous approaches used to lubricate cartilage using brush or bottle type linear polymers. We have added two additional references related to large, non-brush or bottle type, linear polymers [DOI: 10.1529/biophysj.106.088799 (*Biophys. J.* **2007**, *92*, 1693), DOI: 10.1002/jor.23370 (*J. Orthop. Res.* **2017**, *35*, 548), DOI:10.1016/j.cocis.2010.07.002 (*Curr. Opin. Colloid Interface Sci.* **2010**, *15*, 406)] to further complete the list. Additionally, we have added Table 5 to the supporting information listing the previous lubricant system investigated and the determined COF. We acknowledge that it is important to recognize this previous work on other lubricious polymers. However, given the different technical methods (geometry, protocol, load, sliding rate) used to characterize the different lubricants, it is challenging to make direct comparisons between the different polymers. Going forward it will be important as a community to establish an ASTM protocol or similar standardized test for cartilage lubrication so that comparisons can be more easily made. The large globular macromolecules described herein have not been previously synthesized or studied as lubricants for hard or soft surfaces. Similarly sized linear polymers (e.g., polyethylene oxide, polyvinyl alcohol) form gels and are not usable as single molecule lubricants for comparison studies.

We have added the following new text comparing the performance of the mega HPGs with the previous published linear and brush polymers on page 5, of the revised manuscript. “The low COF value observed for the 9 MDa mega HPG-3 is interesting and, thus, we compared its performance to other lubricants (see Supplementary Table 5, a tabulation of previous lubricant performances). However, lubrication performance highly depends on measurement geometry and protocol as well as tissue type. Given the above caveats, the COF values of the mega HPGs are roughly an order of magnitude lower than the values reported for some bottle-brush copolymer lubricants between cartilage and glass surfaces [DOI: 10.1002/jor.23370 (J. Orthop. Res. 2017, 35, 548), DOI:10.1016/j.cocis.2010.07.002 (Curr. Opin. Colloid Interface Sci. 2010, 15, 406)] while on par with linear polyelectrolyte polymers between cartilage and cartilage surfaces [DOI: 10.1021/acsbiomaterials.9b00085 (ACS Biomater. Sci. Eng. 2019, 5, 3060)] (page 6, Paragraph-1).”

Reviewer #3 (Remarks to the Author):

Comment. The paper by Anilkumar et al. describes the synthesis and the characterization of hyperbranched polyglycerols that exhibit a molecular weight up to nearly 10 MDa. The authors can clearly prove that they reached this high molecular weight indeed. Moreover, they can demonstrate that aqueous solutions of these mega-HPGs exhibit low friction and may have interesting medical applications. I believe that this report possesses the sufficient novelty to warrant publication as a communication to Nature. However, there are still some points which should be improved prior to publication:

Response. We thank the reviewer for the positive comments and stating that the results possess sufficient novelty to warrant publication in Nature Communications.

Comment 1. I believe that Figure 12B is very instructive and should be moved to the main text. It should be shown in a double-logarithmic axis. I understand that the intrinsic viscosity should be nearly independent of the molecular weight. However, its value starts to increase considerably for molecular weights beyond 1 MDa. Why is this so? Do the authors have an explanation for this observation?

Response 1. We moved the supplementary figure 12 into main text (now it is Fig. 2) and it is presented in a double logarithmic axis format. We agree with the reviewer that intrinsic viscosity of mega HPG slightly increases with the molecular weight beyond 1 million Dalton. It might be attributing to moderate increase in the size of the mega HPGs [DOI: 10.1126/science.188.4195.1268 (Science 1975, 188, 1268)] after 1 million Dalton. We are further investigating the internal structure of these single molecules using neutron scattering. We anticipate that this will provide more information on why intrinsic viscosity of these molecules are increasing more rapidly beyond 1 MDa.

Figure 6. Comparison of solution properties of mega HPGs (including low molecular weight HPGs, first four data points) with PEG and PAMAM dendrimers. **a.** Variation in hydrodynamic diameter of the polymers with molecular weight. *Mega* HPGs and their low molecular weight counterparts, and PAMAM dendrimers are compact compared to PEG polymers. The values for HPGs (76.5, 307, and 771 kDa) are obtained from literature [DOI: 10.1021/ma0613483 (*Macromolecules* **2006**, 39, 7708)]. For PEGs, the size of PEG-4 and 11 million Dalton was derived from R_g and simulation studies [DOI: S0032-3861 (96)00859-2 (*Polymer* **1997**, 38, 2885)]. **b.** Dependence of intrinsic viscosity of the polymers with molecular weight (arrow shows the representative axis). *Mega* HPGs showed almost similar intrinsic viscosity behavior as PAMAM dendrimers, however, a slight increment in viscosity with molecular weight of mega HPG was observed. The PEG systems showed linear increase in viscosity with molecular weight.

Comment 2. My main problem with the paper is related to the description of the lubrication properties of the aqueous solutions. To my understanding, the excellent lubrication of these solutions is simply due to the low intrinsic viscosity in combination of a strong dependence on shear rate. Is this so? I guess that most of the ordinary readers of this journal are not familiar with Hersey numbers and Stribeck curves. Therefore, I suggest the authors show ordinary flow curves in which the shear viscosity is plotted against the shear rate. This information should already be available from the measurements done so far and would be far more instructive than the present figure 2. What is the physical reason for the low friction? The authors may find it revealing to study the respective investigations of J. Klein and coworkers who analyzed the friction between polymeric surfaces in detail. In turn, Figure 3 could be easily moved into the SI. I find the discussion of the various data gathered in this figure not very instructive and near to incomprehensible.

We have separated the above comment into multiple comments for responding.

Comment 2a. To my understanding, the excellent lubrication of these solutions is simply due to the low intrinsic viscosity in combination of a strong dependence on shear rate. Is this so? Therefore, I suggest the authors show ordinary flow curves in which the shear viscosity is plotted against the shear rate. This information should already be available from the measurements done so far and would be far more instructive than the present figure 2.

Response 2a. We thank the reviewer for the comment and the suggestion. We have added the flow curves to the SI (Supplementary Fig. 12). The viscosity is not dependent on the shear rate for the 7 wt% 1, 3, and 9 MDa polymer solutions, and these compositions behave as Newtonian fluids. Whereas the solution of 1 MDa at 23% demonstrates shear thinning behavior mimicking that of Synvisc although with a smaller change in viscosity as a function of shear rate. The 23 wt% solution of 3 and 9 MDa polymers behave as Newtonian fluids. We surmise that the 3 and 9 MDa polymers are denser than the 1MDa polymer and do not entangle with each other. Additional studies will be needed to support this proposal. The 1 MDa polymer extends into solution, shear thins, and is a non-Newtonian fluid. The hydrodynamic diameter of 9 MDa polymer is only twice the size of the 1 MDa polymer. The 9 MDa polymer maintains a constant viscosity across shear rate. A Newtonian fluid lubricant is advantageous over a non-Newtonian one because the viscosity of the non-Newtonian lubricant reduces as a function of shear displacing it from the surface and increasing the contact area between the surfaces.

We have added the flow curves to the SI (Supplementary Fig. 12) and following discussion to the manuscript on page 6 (line 14) to address the new data. “Analysis of the flow data (Supplementary Fig. 12) reveals only the solution of 1 MDa at 23% shear thins mimicking that of Synvisc, although with a smaller change in viscosity as a function of shear rate. We surmise that the 3 and 9 MDa polymers are denser than the 1MDa polymer. The hydrodynamic diameter of the 9 MDa polymer, for example, is only twice the size of the 1 MDa polymer, and therefore is less likely to entangle with itself. The 1 MDa polymer extends into solution, shear thins, and is a non-Newtonian fluid. The 9 MDa polymer maintains a constant viscosity across shear rate and acts as a Newtonian fluid. A Newtonian fluid lubricant is advantageous over a non-Newtonian one because the viscosity of the non-Newtonian lubricant reduces as a function of shear, displacing it from the surface and increasing the contact area between the surfaces.”

Figure 7. The viscosity-shear rate behavior of mega HPGs at two different concentrations (7 and 23 wt%) and compared with Synvisc One (Synvisc), osteoarthritic synovial fluid (OA SF), bovine synovial fluid (BSF), and saline.

Comment 2b. My main problem with the paper is related to the description of the lubrication properties of the aqueous solutions. I guess that most of the ordinary readers of this journal are not familiar with Hersey numbers and Stribeck curves.

Response 2b. The Stribeck theory is fundamental to characterizing and understanding lubrication. We have added text to the SI describing Hersey numbers and a Stribeck curve. We have also added two additional references on lubrication and Stribeck curves to aid the reader. Specifically, we added the following text to the SI (Supplementary Discussion, Page 5) for the Stribeck curve.

“A Stribeck curve plots COF vs Hersey number, the key variable in a Stribeck curve. The Hersey number, a dimensionless number, is derived from multiplying velocity (m/s) with viscosity ($\text{Pa}\cdot\text{s} = \text{N}\cdot\text{s}/\text{m}^2$), and dividing by load per unit length (N/m). The lubrication in presence of a fluid of incompressible materials has been well characterized with respect to specific modes, classically displayed on Stribeck curves, which defines the lubrication modes and is fundamental in understanding lubrication. In order to elucidate the relative contributions of the mega HPGs and other control lubricants to the different modes of lubrication, we determined the Stribeck curves for the metal on metal surface. Given the experimental constraints for the cartilage on cartilage surface experiments, we performed a more limited experiment.”

Comment 2c. "What is the physical reason for the low friction? The authors may find it revealing to study the respective investigations of J. Klein and coworkers who analyzed the friction between polymeric surfaces in detail. In turn, Figure 3 could be easily moved into the SI. I find the discussion of the various data gathered in this figure not very instructive and near to incomprehensible."

Response 2c. We thank the reviewer for this comment. We have edited the text to improve the clarity, updated the Figure 3 (Figure 4 in the revised manuscript) from bar plot to box plot to better display the groups, and included additional refs including the Klein et al. Additional experiments are needed to fully determine the mechanism of lubrication on cartilage. We hypothesize a few modes of lubrication may be in effect during lubrication with mega HPGs, particularly hydration shell lubrication described by Klein et al., [DOI: 10.1038/s41467-017-01421-7 (Nat. Commun. 2017, 8, 1546), DOI:10.1038/nature05196 (Nature 2006, 444, 191), DOI:10.1038/nature01970 (Nature 2003, 425, 163)], where our highly hydrated, water dense structures of the mega HPGs maintain a molecular water film at the cartilage surface, supporting heavy loads without being squeezed out while simultaneously rapidly relaxing. Additionally, we suspect the 9 million Dalton mega HPG-3 is better retained on the tissue surface and does not get washed off, providing constant and lower COF compared to the PBS, or other lubricants. This proposal is supported by experiments that show no shear thinning with the mega HPG-3 and follows the scenario in which Greene suggests [DOI: 10.1073/pnas.1101002108 (PNAS 2011, 1)] that a "mechanical trapping" mechanism maintains a layer of immobilized HA between surfaces. This is similar to the 'ultra-filtration' hypothesis from Walker et al., [DOI: <http://dx.doi.org/10.1136/ard.27.6.512> (Ann. rheum. Dis. 1968, 27, 512)], where water preferentially flows into the articular surface through the ~10nm bovine cartilage pores [DOI: 10.1016/j.micromeso.2017.01.005 (Microporous and Mesoporous Mater. 2017, 241, 238)], leaving larger molecules, such as mega HPGs with diameters of 20 – 40nm, to aggregate at the leading edge of contact. How these proposed mechanisms relate to the mode of lubrication – whether boundary, mixed, or hydrodynamic – on cartilage, requires further investigation (Main text, page 6).

For the experiments performed on cartilage, we did not vary the velocity, as in the stainless-steel experiments. Thus, the rotation method used here for cartilage was not characterized with respect to the different lubrication regimes. The results are interesting, and the first studies of these novel mega macromolecules provide impetus for continued investigation into the mechanism of lubrication on soft tissues and this new class of polymers. We have edited the main text to include the above information.

We have edited the text, added more literature for comparison and softened our conclusion regarding the lubrication mechanism of these interesting and very new single molecule lubricants:

Discussion, Page 6, lines: 33:

"We propose that the mega HPGs, specifically the 3 and 9 MDa Newtonian fluid lubricants, function as interposed molecular ball-bearings in water to reduce the COF between the stainless-steel surfaces. We hypothesize a few modes of lubrication may be in effect during lubrication of cartilage with mega HPGs, particularly hydration shell lubrication described by Klein et al., [DOI: 10.1038/s41467-017-01421-7 (Nat. Commun. 2017, 8, 1546), DOI:10.1038/nature05196 (Nature 2006, 444, 191), DOI:10.1038/nature01970 (Nature 2003, 425, 163)], where our highly hydrated, water dense structures of the mega HPGs maintain a molecular water film at the cartilage surface, supporting heavy loads without being squeezed out while simultaneously rapidly relaxing [DOI: 10.1021/nn5062707 (ACS Nano 2015, 9, 2614)]. Additionally, we suspect the 9 million Dalton mega HPG-3 is better retained on the tissue surface and does not get washed off, providing constant and lower COF compared to PBS, or other lubricants. This proposal is supported by experiments that show no shear thinning with the mega HPG-3 as it is a Newtonian lubricant and follows the scenario in which Greene suggests [DOI: 10.1073/pnas.1101002108 (PNAS 2011, 1)] that a "mechanical trapping" mechanism maintains a layer of immobilized HA between surfaces. This is similar to the 'ultra-filtration' hypothesis from Walker et al., [DOI: <http://dx.doi.org/10.1136/ard.27.6.512> (Ann. rheum. Dis. 1968, 27, 512)] where water preferentially flows into the articular surface through the ~10 nm bovine cartilage pores leaving larger molecules [DOI: 10.1016/j.micromeso.2017.01.005 (Microporous and Mesoporous Mater. 2017, 241, 238)], such as the 20 – 40 nm diameter mega HPGs to aggregate at the leading edge of contact. How these proposed mechanisms relate to the mode of lubrication on cartilage – boundary, mixed, or hydrodynamic – requires further investigation.

Comment 3. Figure 11A: The cryo-TEM looks as if the systems would be hollow spheres. Could the authors comment on this observation?

Response 3. We thank the reviewer for this observation. Formation of hollow spheres is well known with polymers and dendrimers, for instance, polyethylene [DOI: 10.1002/APP.43207 (J. Appl. Polym. Sci. 2016, 133, 43207)], poly(propyleneimine) dendrimers [DOI: 10.1002/chem.201502852 (Chem. Eur. J. 2015, 21, 18623–18630)], and phenyl ether dendrimers [DOI: 10.1021/ja8034703 (J. Am. Chem. Soc. 2008, 130, 13079)]. However, most of these systems formed by self-assembly process. In our case, the observed structures are single molecule covalent structures rather than formed by the self-assembly process of small molecules. The sizes of these single molecule structures determined by dynamic light scattering and cryo-TEM are similar. To further validate this structural morphology, as a part of future studies, we are currently conducting small-angle neutron scattering (SANS) experiments in collaboration with Professor Dr. Matthias Ballauff, Department of Physics, Humboldt University, Berlin. We anticipate that this will reveal the internal structures of these new single molecule mega macromolecules.

REVIEWERS' COMMENTS:

Reviewer #1 (Remarks to the Author):

The authors have adequately addressed this reviewer's comments and concerns.

Reviewer #2 (Remarks to the Author):

I think the authors have fully addressed my concerned and now it is suitable for publication in Nature communication.

Reviewer #3 (Remarks to the Author):

All comments have been addressed carefully and I recommend the publication of this manuscript as is